Prediabetes in adult Saudis: a systematic review & meta-analysis of prevalence studies (2000–2024)

Mohieldein Abdelmarouf fky@qu.edu.sa 1
Baig Habeeb Ali 2
Elhabiby Mahmoud 3
Almushawwah Abdullah 1
Mahfouz Mohamed Salih 4
Khirelsied Atif H. 5
Abdelmarouf Nour 6
Modawe GadAllah 7
1 Department of Medical Laboratories, College of Applied Medical Sciences, Qassim University , Buraidah , Qassim , Saudi Arabia
2 Department of Microbiology, College of Medicine, Northern Border University , Arar , Saudi Arabia
3 Department of Medical Laboratory Sciences, Al-Aqsa University , Gaza , Palestine
4 Department of Family and Community Medicine, Faculty of Medicine, Jazan University , Jazan City , Jazan , Saudi Arabia
5 Department of Biochemistry, Faculty of Medicine, Al-Baha University , Al Baha , Saudi Arabia
6 Faculty of Medicine, Alexandria University , Alexandria , Egypt
7 Faculty of Medicine and Health Sciences, Department of Biochemistry, Omdurman Islamic University , Omdurman , Sudan
Menini Stefano
Electronic publication date: 2025 Aug 20
Publication date: 2025
Volume: 13
Electronic Location ID: e19778
Received 2024 Dec 6; Accepted 2025 Jun 30
Copyright: ©2025 Mohieldein et al.
Copyright year: 2025
Copyright holder: Mohieldein et al.
License: This is an open access article distributed under the terms of the Creative Commons Attribution License, which permits unrestricted use, distribution, reproduction and adaptation in any medium and for any purpose provided that it is properly attributed. For attribution, the original author(s), title, publication source (PeerJ) and either DOI or URL of the article must be cited.
License URL: https://creativecommons.org/licenses/by/4.0/

Keywords: Prediabetes, Cross-sectional, Saudi Arabia, Adults, Lifestyle

Funding: The authors received no funding for this work.

==============================
Background

There is a lack of national data on the pooled prevalence of prediabetes among adult Saudis. This study aimed to provide a comprehensive estimate of the prevalence of prediabetes among Saudi adults.

Methods

We searched databases for cross-sectional studies conducted between January 2000 and September 2024. We included studies written in English and conducted in Saudi Arabia. The studies had to report the prevalence of prediabetes among adults (≥18 years) using American Diabetes Association (ADA) or the World Health Organization (WHO) criteria. We excluded studies that involved non-adult or non-Saudi populations. We also excluded studies published before 2000 or those without clear diagnostic criteria or prevalence data. The review followed the Preferred Reporting Items for Systematic Reviews and Meta-Analyses Statement guidelines. Pooled prevalence was calculated using a random-effects model. Subgroup and sensitivity analyses were performed. We assessed the quality of the included studies using the Joanna Briggs Institute (JBI) checklist for prevalence studies. We used Comprehensive Meta-Analysis software version 4.0 to perform the statistical calculations.

Results

Eighteen cross-sectional studies were analyzed, including 47,718 adult Saudis from the general population. The pooled national prevalence of prediabetes among Saudi adults was estimated at 24.1% (95% CI [19.5%–29.4%]). Sensitivity analysis confirmed this pooled prevalence. Funnel plot inspection and Egger’s test indicated no substantial publication bias. Subgroup analyses revealed a higher prevalence in fasting blood glucose (FBG) studies compared to glycated hemoglobin (HbA1c) studies: 25.7% (95% CI [16.6%–37.6%]) vs. 23.7% (95% CI [16.7%–32.5%]); males had a higher prevalence than females 34.8% (95% CI [25.4%–45.5%]) vs. 18.7% (95% CI [10.9%–30.2%]). Meta-regression analysis indicated a slight upward trend in prevalence over time, with a positive coefficient for ‘Year’ (0.027).

Discussion

A high pooled prevalence of prediabetes was documented among the adult Saudi population, with a markedly higher prevalence among males. These findings emphasize the need for early lifestyle interventions, optimized screening programs, and effective resource allocation to prevent the progression to type 2 diabetes. We acknowledge the high level of heterogeneity among the included studies. Additionally, we note that no eligible studies specifically from the northern region of Saudi Arabia were included in the meta-analysis.

Introduction

Diabetes is becoming increasingly prevalent at an alarming rate. This trend is leading to worsening impacts on individuals, the economy, and society as a whole (Perveen et al., 2016). The healthcare community is showing a growing focus on the prevention and prediction of diabetes (Cho et al., 2018). Recent data from the International Diabetes Federation (IDF) indicate that 415 million adults worldwide have been diagnosed with diabetes, with an additional 318 million individuals experiencing impaired glucose tolerance (Dawish & Robert, 2021). Additionally, these numbers are projected to rise to 642 million for diabetes and 482 million for impaired glucose tolerance by 2040 (Cefalu et al., 2016).

Prediabetes, also known as impaired fasting glucose or impaired glucose tolerance, is a condition in which blood glucose levels are higher than normal but have not yet reached the cutoff for diabetes (Rett & Gottwald-Hostalek, 2019; Amelia et al., 2023). Individuals with prediabetes are at risk of developing diabetes and experiencing both microvascular and macrovascular complications (Nathan et al., 2007; Brannick & Dagogo-Jack, 2018). The Diabetes Prevention Program reported that, on average, about 11% of individuals with prediabetes develop type 2 diabetes each year over a three-year follow-up period (Al-Zahrani et al., 2019). Consequently, timely intervention and management of prediabetes are crucial to prevent its progression to diabetes (Gottwald-Hostalek & Gwilt, 2022). Findings from several randomized controlled studies have shown that individuals at high risk of developing type 2 diabetes can successfully prevent its progression. This success is attributed to early, intensive interventions that are tailored to each individual’s unique needs (Li et al., 2008; Ramachandran et al., 2006; Lindström et al., 2006; Kosaka, Noda & Kuzuya, 2005).

Saudi Arabia has undergone significant lifestyle changes in recent years. Rapid urbanization, shifts in dietary patterns, and increased sedentary behavior are among the most notable changes. Together, these factors have contributed to the rising risk of prediabetes and diabetes (Munawir Alhejely et al., 2023). Both prediabetes and type 2 diabetes are common in Saudi Arabia and have shown a rising trend over time (Aldossari et al., 2018). Historical data on prediabetes prevalence in Saudi Arabia provide valuable context for understanding trends over time. A large survey of 13,177 Saudi subjects over 15 years of age reported IGT prevalence rates of 10% and 8% among urban and rural males, respectively, and 11% and 8% among urban and rural females (Al-Nuaim, 1997). A regional study conducted in Najran found even lower prevalence, with rates of 1.02% in males and 0.693% in females over 14 years old, with a modest increase among individuals over 30 years (EI-Hazmi et al., 1997). Similarly, a national household survey of 23,493 participants reported IGT prevalence rates of 0.717% in males and 1.347% in females aged 14–70 years (El-Hazmi et al., 1996).

Enhancing healthcare services and reducing the prevalence of chronic diseases are key goals of the healthcare initiatives under Saudi Vision 2030 (Mani & Goniewicz, 2024). Tremendous improvements in healthcare services and the Saudi health system have been recognized. These advancements support efforts in health promotion and disease prevention (Marwa Tuffaha, 2015). Although several studies on prediabetes have been conducted in various regions of Saudi Arabia, no systematic review and meta-analysis (SRMA) has been published to specifically estimate the pooled prevalence of prediabetes among the adult Saudi population. Only one SRMA has assessed prediabetes prevalence in Saudi Arabia, as part of a broader study of Eastern Mediterranean Region (EMRO) countries (Mirahmadizadeh et al., 2020). We believe that a comprehensive SRMA on prediabetes prevalence among Saudi adults is essential for accurately assessing the disease burden. Therefore, we expect that this analysis will strengthen public health interventions aligned with the goals of Saudi Vision 2030. Ultimately, it could help reduce the future burden of diabetes. This SRMA aimed to identify, summarize, and estimate the pooled prevalence of prediabetes among adult Saudis living in Saudi Arabia. The analysis included studies published between January 2000 and September 2024.

Material and Methods

Search strategy

A search strategy was developed to answer the research question, ‘What is the pooled prevalence of prediabetes among the adult Saudi population in Saudi Arabia?’

This strategy was formulated using the Population, Intervention, Comparison, and Outcome (PICO) approach. The population included adult Saudi individuals aged 18 years and older. The intervention or exposure focused on factors associated with prediabetes. Comparison was not applicable, as the study aimed to estimate prevalence rather than compare groups. The outcome was the prevalence of prediabetes based on diagnostic criteria established by the American Diabetes Association (ADA) or the World Health Organization (WHO).

A comprehensive electronic search was conducted from October 10 to 12, 2024, covering studies published between January 2000 and September 2024. The same search dates were applied across all databases, including MEDLINE (PubMed), Web of Science, Google Scholar, ScienceDirect, and the Saudi Digital Library. Additionally, the reference lists of the included studies were reviewed to ensure thorough coverage of the existing literature. Synonyms were systematically identified using the Medical Subject Headings (MeSH) vocabulary. They were incorporated into the search strategy and combined using Boolean operators (OR and AND) to retrieve relevant studies. The following keywords were used: prediabetes, glucose intolerance, impaired fasting glucose, impaired glucose tolerance, prevalence, and Saudi Arabia. We used Zotero software (Corporation for Digital Scholarship, Vienna, VA, USA) to screen the titles and abstracts of search results and ensure the relevance of the studies. This initial screening was followed by full-text screening to identify studies eligible for inclusion in the meta-analysis based on predefined criteria. Title and abstract screening were performed by four authors (H.A.B, M.E., A.A., N.A), while full-text screening was conducted by two authors (A.M., M.S.M.). The literature screening was carried out from October 13 to 16, 2024.

The systematic review was conducted in accordance with the PRISMA (Preferred Reporting Items for Systematic Reviews and Meta-Analyses) Statement (Moher et al., 2010). It was registered in the PROSPERO International Prospective Register of Systematic Reviews (registration number CRD42024600640). Tables S1 and S2 includes the PRISMA checklist and details the search strategy used across the databases.

Inclusion and exclusion of studies

We included studies that met all of the following criteria: primary studies with a cross-sectional design involving adult Saudi citizens (aged ≥ 18 years) residing in Saudi Arabia; studies from any region of Saudi Arabia reporting the prevalence of prediabetes in the general population based on diagnostic criteria from the American Diabetes Association (ADA) or the World Health Organization (WHO); and studies published in English between January 2000 and September 2024. Additionally, studies were included if they provided both the number of events and sample size to enable recalculation of the required estimates. English is widely used in academic and medical research in Saudi Arabia. As a result, most relevant publications are expected to be in English. This minimizes the risk of missing significant data. The time frame from January 2000 to September 2024 was chosen to ensure the use of consistent, up-to-date diagnostic criteria and to capture the most relevant studies.

Studies were excluded if they met any of the following conditions: study designs such as case-control, case reports, case series, randomized controlled trials, editorials, review articles; articles with incomplete information; or non-primary research; studies involving populations outside Saudi Arabia or mixed populations of Saudi and non-Saudi residents within Saudi Arabia; pediatric or adolescent populations (under 18 years of age); published before 2000; or if they provided inadequate or unclear information on the prevalence and diagnostic criteria for prediabetes.

Data extractions

A data extraction form was developed to collect relevant information from the publications included in this study. Two authors (A.M. and M.S.M.) independently extracted the necessary data on October 17–18, 2024, and double-checked it against the pre-established eligibility criteria. Any conflicts were resolved through consensus between the authors. Although formal inter-rater reliability measures (e.g., Cohen’s kappa) were not calculated, this approach helped ensure accuracy and consistency in the data collected.

The following information was recorded for each study: first author’s last name, year of publication, location, study setting (rural/urban), gender, age, study design, sampling method, number of events, sample size, prevalence of prediabetes, diagnostic criteria (ADA/WHO), and diagnostic tests (fasting blood glucose (FBG), glycated hemoglobin (HbA1c), or mixed).

Operational definitions

The ADA defines prediabetes as an HbA1c level of 5.7%–6.4%, FBG of 100–125 mg/dL, or a 2-hour plasma glucose level of 140–199 mg/dL following a 75 g oral glucose tolerance test. The WHO uses a narrower definition, with FBG levels of 110–125 mg/dL and 2-hour plasma glucose levels of 140–200 mg/dL (Bansal, 2015; Al-Omar et al., 2024).

Both ADA and WHO diagnostic criteria were accepted, and the specific criteria used in each study were carefully recorded. To address differences in diagnostic thresholds, subgroup analyses by test type (FBG, HbA1c, or mixed) were conducted to assess variability and minimize their impact on the pooled prevalence estimate.

Appraisal of the quality of included studies

To assess the quality of the extracted articles, we used the Joanna Briggs Institute (JBI) quality assessment scale for prevalence studies (Munn et al., 2015). Four authors (H.A.B, M.E., A.A., N.A) independently scored the articles, with cross-checking by two additional reviewers (A.M., M.S.M.). Any discrepancies in scoring were resolved through discussion and consensus among the reviewers. Study quality was categorized into three levels: high (8–9), medium (7–6), and low (below 6).

The JBI Critical Appraisal Checklist comprises nine questions that evaluate different dimensions of study quality: (1) the appropriateness of the sample frame for the target population, (2) the adequacy of participants sampling, (3) the sufficiency of sample size, (4) the detail provided regarding study subjects and setting, (5) the comprehensiveness of data analysis concerning the identified sample, (6) the validity of methods used to identify the condition, (7) the consistency and reliability of condition measurement across participants, (8) the appropriateness of statistical analysis performed, and (9) the adequacy of the response rate, or the management of low response rates.

Assessing the overall certainty of evidence

To assess the overall certainty of evidence, we applied the Grading of Recommendations Assessment, Development and Evaluation (GRADE) approach to the pooled prevalence estimate. GRADE evaluates five domains: risk of bias, inconsistency, indirectness, imprecision, and publication bias.

Statistical analyses

Statistical analyses were performed using Comprehensive Meta-Analysis software, version 4.0 (Biostat, Englewood, NJ, USA). The pooled prevalence results were reported as proportions with 95% confidence intervals (CIs) and presented in forest plots. Heterogeneity among the studies was assessed using Cochrane’s Q statistic (Cochran, 1954) and quantified by calculating the I2 (Chen & Benedetti, 2017). Subgroup analyses were conducted to explore potential sources of substantial heterogeneity, specifically across gender, diagnostic tests, quality assessment, and study regions.

A meta-regression analysis was performed to examine the relationship between the year of publication of selected articles and the effect size, specifically the prevalence of prediabetes. Publication bias was evaluated using funnel plots and Egger’s test (Stuck, Rubenstein & Wieland, 1998), with a p-value less than 0.05 indicating statistically significant publication bias. The random-effects model was applied rather than the fixed-effects model, as it accounts for variance between studies and allows for broader generalization of the results.

Results

Study selection

Figure 1 presents the schematic flow of the identification and inclusion processes for the studies. A total of 730 records were identified through the literature search, with 313 duplicates subsequently removed. The remaining 417 publications were screened by title and abstract, and 370 articles were found irrelevant and excluded. The remaining 47 articles were evaluated for eligibility through the full-text screening, and 30 were excluded. The reasons for excluded articles were categorized as population issues (n = 14); study design (n = 3); diagnostic methods (n = 10); outcome (n = 2); and data collection methods (n = 1) (see Table S3). An additional eligible study was identified by examining reference lists from the included studies, resulting in a final total of 18 included studies (Al-Nozha et al., 2004; Mirza et al., 2013; Al-Rubeaan et al., 2015; Ghoraba et al., 2016; Turki, Hegazy & Abaalkhail, 2016; Aldossari et al., 2018; Aldossari et al., 2020; Al-Ghamdi et al., 2018; Aljabri, 2018; Al Amri et al., 2019; Al-Zahrani et al., 2019; Bahijri et al., 2020; Latif & Rafique, 2020; Fayed et al., 2022; Al Shehri et al., 2022; Alomari & Al Hisnah, 2022; Abu-Almakarem, 2024; Alhomaid & Moin Ahmed, 2024). The eighteen studies were conducted across various regions of Saudi Arabia, encompassing a total population of 47,718 adult Saudis.

Figure 1 PRISMA flow diagram.

The selection of the included articles in SRMA on the prevalence of prediabetes in the adult Saudi population (2000–2024).

The regional distribution of the excluded studies by full-text screening was 46.6% (n = 14) for Central region, 20% (n = 6) for Western, 10% (n = 3) for Southern, 10% (n = 3) for Northern, 6.7% (n = 2) for Eastern, and 6.7% (n = 2) for a nationwide study. None of the three studies conducted in the Northern region (Alreshidi et al., 2023; Sebeh Alhazmi et al., 2017; Ali Moustafa Marzok Elkhateeb, 2018) satisfied the eligibility criteria for the meta-analysis.

The characteristics of the included studies

Table 1 presents the summary characteristics of the included studies. Eighteen studies met the inclusion criteria and employed a cross-sectional observational design. Eight studies (44.4%) were conducted in the Central region of Saudi Arabia, one study (5.6%) in the Eastern region, two studies (11.1%) in the Southern region, five studies (27.8%) in the Western region, and two studies (11.1%) were nationwide, encompassing several multiple of Saudi Arabia. Although no separate study from the North region was eligible for this review, it could be incorporated into the two nationwide studies (Al-Nozha et al., 2004; Al-Rubeaan et al., 2015). Studies included in this meta-analysis were published between 2004 and 2024. Sixteen studies were graded as “high” quality, while two studies were graded as “medium” quality, based on the JBI Critical Appraisal Checklist for analytical cross-sectional studies, see Table S4.

Table 1 Baseline characteristics of selected studies in the systematic review and meta-analysis on the prevalence of prediabetes in the adult Saudi population (2000–2024).

	Author, Publication year	Region of study	Setting Rural/Urban	Nationality	Age	Gender	Study design	Sampling method	Number of cases	Sample size	Prevalence of prediabetes	Diagnostic criteria ADA vs. WHO	Diagnostic test	
1	Abu-Almakarem, 2024	Al-Baha	Urban	Saudi	20–57	Male	Cross-sectional	NA	40	165	24.24%	ADA	A1c	
2	Al Amri et al., 2019	Jeddah	Urban	Saudi	≥18	Mixed	Cross-sectional	Stratified two-stage cluster sampling	176	613	28.7%	ADA	Mixed	
3	Al Shehri et al., 2022	Riyadh	Urban	Saudi	18–30	Male	Cross-sectional	Random selection	1,110	2,010	55.2%	ADA	FBG	
4	Aldossari et al., 2018	Al-Kharj	Urban	Saudi	18–60	Male	Cross-sectional	Multi-stage cluster sampling	105	381	27.56%	ADA	A1c	
5	Aldossari et al., 2020	Al-Kharj	Urban and rural	Saudi	≥18	Mixed	Cross-sectional	Multi-stage cluster sampling	231	1,019	22.67%	ADA	A1c	
6	Aljabri, 2018	Jeddah	Urban	Saudi	≥20	Mixed	Cross-sectional	Random sampling	362	1,095	33.06%	ADA	A1c	
7	Alomari & Al Hisnah, 2022	Al Bahah	Urban	Saudi	30–50	Mixed	Cross-sectional	Simple random sampling	76	378	20.1%	ADA	Mixed	
8	Al-Rubeaan et al., 2015	Nationwide	Both urban and rural	Saudi	≥30	Mixed	Cross-sectional	Random selection	4,599	18,034	25.5%	ADA	FBG	
9	Al-Zahrani et al., 2019	Alkharj	Urban	Saudi	18–60	Female	Cross-sectional	Multi-stage stratified cluster sampling	120	638	18.8%	ADA	A1c	
10	Bahijri et al., 2020	Jeddah	Urban	Saudi	≥20	Mixed	Cross-sectional	Stratified two-stage cluster sampling	259	1,403	18.46%	ADA	Mixed	
11	Fayed et al., 2022	Riyadh	Urban	Saudi	≥18	Mixed	Cross-sectional study	WHO-STEPwise approach	620	2,470	25.1%	ADA	A1c	
12	Latif & Rafique, 2020	Dammam	Urban	Saudi	18–20	Female	Cross-sectional	Random sampling	56	300	18.7%	ADA	FBG	
13	Mirza et al., 2013	Makkah	Urban	Saudi	≥20	Male	Cross-sectional	Random sampling	46	141	32.7%	ADA	Mixed	
14	Turki, Hegazy & Abaalkhail, 2016	Makkah	Urban	Saudi	40-70	Mixed	Cross-sectional	random sampling	39	225	17.3%	ADA	A1c	
15	Ghoraba et al., 2016	Riyadh	Urban	Saudi	18–62	Mixed	Cross-sectional	Convenience sampling	120	510	23.6%	ADA	FBG	
16	Al-Ghamdi et al., 2018	Al-Kharj	Mixed	Saudi	18-67	Mixed	Cross-sectional	Multi-stage random sampling	231	1,019	22.7%	ADA	A1c	
17	Alhomaid & Moin Ahmed, 2024	Qassim	Urban	Saudi	18–35	Mixed	Cross-sectional	Random sampling	76	400	19%	ADA	Mixed	
18	Al-Nozha et al., 2004	Nationwide	Both urban and rural	Saudi	30–70	Mixed	Cross-sectional	Stratified cluster sampling	2,388	16,917	14.1%	ADA	FBG	

Pooled national prevalence of prediabetes among adult Saudis

A total of 18 eligible reports were included in the systematic review, with an overall sample size of 47,718. The pooled national prevalence of prediabetes in the Saudi adult population, as determined by WHO/ADA criteria, was estimated to be 24.1% (95% CI [19.5%–29.4%]). This estimate was calculated using the random-effects model, as shown in Fig. 2. This prevalence indicates that a significant portion of the Saudi adult population is affected; that is, approximately one in four adults in Saudi Arabia may be prediabetic.

Figure 2 Forest plot displays the mean effect size.

The pooled prevalence estimate (the black diamond) was around 24.1% (95% CI [19.5%–29.4%]) for a sample of 18 studies. Most of the studies show relatively narrow confidence intervals, suggesting robust estimates within those individual samples. Sources: Al-Nozha et al., 2004; Mirza et al., 2013; Al-Rubeaan et al., 2015; Ghoraba et al., 2016; Turki, Hegazy & Abaalkhail, 2016; Aldossari et al., 2018; Aldossari et al., 2020; Al-Ghamdi et al., 2018; Aljabri, 2018; Al Amri et al., 2019; Al-Zahrani et al., 2019; Bahijri et al., 2020; Latif & Rafique, 2020; Fayed et al., 2022; Al Shehri et al., 2022; Alomari & Al Hisnah, 2022; Abu-Almakarem, 2024; Alhomaid & Moin Ahmed, 2024.

A sensitivity analysis was conducted by systematically removing each study from the meta-analysis to assess the robustness of the pooled estimate. Findings from the sensitivity test indicated that the pooled prevalence of prediabetes remained stable at 24.1% (95% CI [19.5%–29.4%]), regardless of which study was removed. The lower and upper confidence limits also remained relatively stable across the sensitivity analysis. Furthermore, excluding the study by Al Shehri et al. (2022) which had one of the highest individual prevalence rates, only slightly reduced the pooled prevalence to 22.7% (95% CI [19.4%–26.3%)]. Figure S1 illustrates the findings from the sensitivity analysis.

Publication bias assessment

We initially assessed potential publication bias by the funnel plot, which plots the standard error against the logit event rate. Visual inspection of the funnel plot revealed no significant asymmetry, indicating no significant publication bias among the included studies. Instead, the selected studies showed symmetrical distribution around the pooled effect size in the funnel shape (Fig. S2). Egger’s test further confirmed this, with an intercept of 1.97 (95% CI [−6.25–10.19]) and a non-significant two-tailed p-value of 0.618. A p-value >0.05 indicates no evidence of publication bias, consistent with the funnel plot’s visual findings. This suggests that publication bias is not a major concern in this meta-analysis (Tables S5).

Heterogeneity across studies

The heterogeneity analysis showed a high I2 value of 99.1%, indicating substantial heterogeneity across the selected studies. The Cochrane Q-value of 1,920.25 with 17 degrees of freedom is statistically significant (p < 0.001), confirming considerable variability in the prevalence estimates. Therefore, we decided from the start to apply a random-effects model rather than a fixed-effects model, as it accounts for variability among studies and enables generalizability of the results. Using this approach, the pooled prevalence was estimated at 24.1% (95% CI [19.5%–29.4%]) (Tables S6).

Moreover, the high heterogeneity highlighted the need to explore potential sources of variability. Therefore, we conducted subgroup analyses based on (1) the region of the study, (2) diagnostic criteria (A1c vs. FBG), and (3) gender.

Sub-group analyses

Data analyses revealed that studies using FBG as the diagnostic criterion reported the highest prevalence of prediabetes (Fig. 3). The pooled prevalence of prediabetes was 25.7% (95% CI [16.6%–37.6%]) for FBG; 23.7% (95% CI [16.7%–32.5%]) for HbA1c; and 23.3% (95% CI [14.8%–34.6%]) for studies using mixed diagnostic methods (including both HbA1c and FBG).

Figure 3 Forest plot findings based on subgroup analysis of diagnostic Criteria.

HbA1c (glycated hemoglobin), FBG (fasting blood glucose), mixed i.e., studies using a combination of diagnostic tests (A1c, FBG). Sources: Al-Nozha et al., 2004; Mirza et al., 2013; Al-Rubeaan et al., 2015; Ghoraba et al., 2016; Turki, Hegazy & Abaalkhail, 2016; Aldossari et al., 2018; Aldossari et al., 2020; Al-Ghamdi et al., 2018; Aljabri, 2018; Al Amri et al., 2019; Al-Zahrani et al., 2019; Bahijri et al., 2020; Latif & Rafique, 2020; Fayed et al., 2022; Al Shehri et al., 2022; Alomari & Al Hisnah, 2022; Abu-Almakarem, 2024; Alhomaid & Moin Ahmed, 2024.

Regarding gender-based subgroup analysis, studies involving male-only populations (n = 4) had the highest pooled prevalence of prediabetes (Fig. S3). The pooled prevalence was 34.8% (95% CI [25.4%–45.5%]) for male-only studies, 18.7% (95% CI [10.9%–30.2%]) for female-only studies (n = 2), and 22.2% (95% CI [18.1%–26.8%]) for mixed-gender populations (n = 12). A sensitivity analysis for male-only studies was conducted, excluding the Al Shehri et al. (2022) study. The results showed that the pooled prevalence decreased to 27.9% (95% CI [24.1%–32.1%]) but still remained the highest among the subgroups (Fig. S4).

The region-specific subgroup analysis revealed geographical variations in prediabetes prevalence across Saudi Arabia (Fig. S5). The Central region had the highest pooled prevalence at 26.0% (95% CI [19.0%–34.5%]), followed closely by the Western region at 25.4% (95% CI [16.9%–36.4%]). In contrast, the Southern region reported a lower prevalence of 22.1% (95% CI [11.0%–39.3%]), and the Eastern region had the lowest prevalence at 18.7% (95% CI [6.7%–42.5%]). A sensitivity test was conducted by excluding the Al Shehri et al. (2022) study, which reported a Central region prevalence of 55%. This exclusion reduced the pooled prevalence for the Central region to 22.8% (95% CI [20.8%–24.9%]) (Fig. S6).

We conducted a sensitivity analysis based on the quality of the included studies, categorized as either high or medium quality according to the Joanna Briggs Institute (JBI) assessment tool. The pooled prevalence from high-quality studies (n = 16) was 24.2% (95% CI [19.3%–29.8%]), while the pooled prevalence from medium-quality studies (n = 2) was 23.9% (95% CI [12.1%–41.8%]). These findings suggest that the overall pooled estimate remains consistent regardless of study quality, indicating that study quality did not substantially influence the observed prevalence of prediabetes (Fig. S7).

Results of the meta-regression

The meta-regression analysis indicated a slight upward trend in prediabetes prevalence over time (Fig. 4), with the regression equation −55.633 + (0.027 × Year) and a positive coefficient for ‘Year’ of 0.027. However, with a p-value of 0.2312 (95% CI [−0.0172–0.0712]), this relationship was not statistically significant.

Figure 4 Meta-regression of logit event rate (prevalence of prediabetes) vs. year of publication.

The studies Bahijri et al. (2020) and Al-Zahrani et al. (2019) were superimposed in one circle due to relatively similar prevalence rates.

According to the GRADE approach, the certainty was rated as moderate, with a downgrade for inconsistency due to high heterogeneity (I2 = 99.1%). Other domains, including risk of bias, indirectness, imprecision, and publication bias, were considered not serious (Table S7).

Secondary outcomes

Multiple logistic regression analysis from Aljaadi et al. (2023) revealed that the weekly consumption of soft and energy drinks in Saudi adults (n = 3,928) was significantly associated with several sociodemographic and behavioral factors. Specifically, higher consumption was linked to male gender, younger age (<30 years), lower monthly income, and lower levels of physical activity. Moreover, individuals with overweight or obesity were less likely to consume energy drinks than those with healthy weight. However, education level was not associated with either soft or energy drink consumption (Tables S8A & S8B).

Discussion

Early detection and treatment of prediabetes can prevent diabetes, reduce healthcare costs, and lead to significant improvements in public health. A clearer understanding of the pooled prevalence of prediabetes is essential for assessing the effectiveness of intervention strategies and guiding future prevention efforts (Tuso, 2014). In this meta-analysis, we synthesized studies from the past 24 years to address knowledge gaps and provide a comprehensive estimate of prediabetes prevalence among adult Saudis in Saudi Arabia. Additionally, this study examined variations in prediabetes prevalence based on diagnostic methods, gender, study quality, and regional distribution. The pooled prevalence of prediabetes across all eligible cross-sectional studies (n = 18) was 24.1% (95% CI [19.5%–29.4%]). Moreover, the sensitivity analysis showed that the pooled prevalence of prediabetes remained stable at 24.1% (95% CI [19.5%–29.4%]), regardless of which study was removed. These results suggest that no single study or outlier significantly skews the overall observed prevalence, supporting the reliability and robustness of the pooled national estimate of prediabetes among adult Saudis. This high prevalence indicates a substantial prediabetic burden among the adult Saudi population, underscoring a serious public health concern.

Although most studies showed relatively narrow confidence intervals, a few—such as Al Shehri et al. (2022)—exhibited notably wide intervals. This may be attributed to factors such as small sample sizes, population-specific characteristics (e.g., young males applying for military colleges), and potential selection bias. While these studies contributed to the pooled estimate, sensitivity analysis confirmed that their influence does not disproportionately affect the overall pooled prevalence.

Worldwide, the pooled prevalence of prediabetes has been reported across several populations. In the Eastern Mediterranean Region (EMRO), Mirahmadizadeh et al. (2020) reported an overall pooled prevalence of prediabetes of 12.78% (95% CI [10.67–14.89]%). Additionally, the authors reported a prevalence of 14% (95% CI [4.9%–23.1%]) for Saudi Arabia based on seven studies. The discrepancy in the pooled prevalence of prediabetes in Saudi Arabia between the two meta-analyses may be explained by the following factors. Of the seven studies included in the Mirahmadizadeh et al. meta-analysis, only two (Al-Nozha et al., 2004; Al-Rubeaan et al., 2015) met our eligibility criteria and were included in the current meta-analysis. Three studies were excluded due to population issues (El Bcheraoui et al., 2014; Bahijri et al., 2016; Al, Al Osaimi & AL-Gelban, 2007). A fourth study was excluded because of unclear diagnostic methods for prediabetes (Al Baghli et al., 2010); and a fifth study was excluded due to an irrelevant the outcome (Memish et al., 2015). Details on the reasons for excluding studies during full-text screening are provided in Table S3. In a meta-analysis of the adult Cameroonian population residing in Cameroon, Bigna et al. (2018) reported a pooled prevalence of prediabetes of 7.1% (95% CI [3.0%–21.9%]) based on four studies with a combined sample size of 5,872 individuals. In another meta-analysis of the Bangladeshi population, Akhtar et al. (2020) reported a pooled prevalence of prediabetes of 10.1%, based on a combined sample size of 56,452 individuals across 17 studies. Additionally, among populations with similar socioeconomic characteristics, the pooled prevalence of prediabetes was 11.62% (95% CI [7.17%–16.97%]), based on nine studies involving 88,702 individuals from Malaysia (Akhtar et al., 2022). Furthermore, the pooled prevalence of prediabetes in the East African population (n = 43,379) across six countries—Uganda, Tanzania, Ethiopia, Kenya, Rwanda, and Sudan—was 12.58% (95% CI [10.30%–14.86%]) (Asmelash et al., 2023). A separate meta-analysis of the Ethiopian population reported a pooled prevalence of impaired fasting glucose at 8.94%, with a 95% confidence interval (CI) of [2.60%–15.28%] (Yitbarek et al., 2021). A summary of the pooled prevalence of prediabetes across different populations is presented in Tables S9. The Saudi population has the highest prevalence rate, as demonstrated by the comparative analysis of pooled prediabetes prevalence across the populations mentioned above. This may be attributed to increased caloric intake, reduced physical activity, and lifestyle changes linked to urbanization in Saudi Arabia and the Gulf countries overall. A systematic review of 65 studies (Al-Hazzaa, 2018) found that most Saudis do not meet the recommended levels of moderate to vigorous physical activity. The authors identified several barriers to physical activity, including heavy traffic, extreme weather, cultural constraints, limited social support, and lack of time. The Saudi Food and Drug Authority (SFDA)(2024) mandated the display of calorie information on menus in restaurants, hotels, and cafes. This initiative aims to promote health in Saudi Arabia by encouraging healthier food choices in line with international public health standards. Despite these efforts, a recent systematic review found that most Saudis do not adhere to dietary guidelines. Many, especially students and young adults, prefer fast food and processed foods because of their convenience and accessibility (Almutairi et al., 2023). In addition, lack of willpower, limited time, and insufficient social support have been identified as the primary barriers to following a healthy diet (Aldosari et al., 2024). A cross-sectional study also revealed that one-third of adult participants (59.3% male) reported sleeping less than the recommended seven hours per night. This finding highlights the high prevalence of short sleep duration among Saudi adults (Ahmed et al., 2017). Bruckner et al. (2024) emphasized the importance of lifestyle factors in managing insulin resistance and inflammation. They concluded that lower adherence to health-promoting lifestyle habits is associated with increased insulin resistance. A meta-analysis of 30 cohort comparisons reported that strict adherence to multiple low-risk lifestyle behaviors significantly reduces the risk of developing type 2 diabetes by 85%. These behaviors include maintaining a healthy body weight, following a healthy diet, engaging in regular physical activity, and avoiding smoking (Khan et al., 2023).

The high pooled prevalence of prediabetes (24.1%) found in our meta-analysis is consistent with Saudi Arabia’s status as one of the countries with the highest diabetes prevalence worldwide. According to the International Diabetes Federation (IDF), the age-adjusted prevalence of diabetes among Saudi adults (aged 20–79 years) was estimated at 23.1% in 2024, significantly higher than the global average of 11.1% (IDF DIABETES ATLAS, 2015). These findings support the hypothesis that lifestyle factors are key contributors to this burden.

The high heterogeneity (I2 = 99.1%) observed in this meta-analysis necessitates cautious interpretation of the pooled prevalence estimate. Several factors may explain for this variability, including regional lifestyle differences, variations in sampling strategies—such as convenience sampling or recruitment from specific subpopulations—which may introduce selection bias, and inconsistencies in diagnostic methods (FBG vs. HbA1c) and criteria (ADA vs. WHO) used across studies. To explore these sources of heterogeneity, we conducted subgroup analyses based on the region of the study, diagnostic criteria (A1c vs. FBG), gender, and quality of the included studies.

Subgroup analysis revealed higher prevalence estimates in studies using FBG (25.7% (95% CI [16.6%–37.6%]) compared to those using HbA1c (23.7%; 95% CI [16.7%–32.5%]) and mixed methods (23.3%; 95% CI [14.8%–34.6%]). The variation in prediabetes prevalence between studies using FBG and HbA1c reflects important methodological differences. These differences may influence prevalence estimates and contribute to the high heterogeneity observed in this meta-analysis. The higher prevalence observed in studies using FBG (n = 5) studies suggests that the FBG method may be more sensitive and capture more individuals in the prediabetic range than HbA1c (n = 8). Moreover, this discrepancy highlights the importance of standardizing diagnostic methods in epidemiological research, as inconsistencies can contribute to variability in prevalence estimates. In this meta-analysis, significant variability was observed among studies (I2 = 99.1%, p < 0.001). Consequently, we applied a random-effects model instead of a fixed-effects model, which yielded a pooled prevalence of 24.1% (95% CI [19.5%–29.4%]). From this perspective, Ghazanfari et al. (2010) reported that FBS was a more accurate predictor of HbA1c than HbA1c was of FBS. Nomura, Inoue & Akimoto (2012) reported that FPG was generally the strongest predictor of type 2 diabetes in the general population. However, among individuals with FPG levels between 6.1–6.9 mmol/L, its predictive power was comparable to that of HbA1c in men, while HbA1c was more effective in women. The authors proposed a two-step screening approach that combines HbA1c and FPG to improve the accuracy of diabetes prediction. The majority of studies in this meta-analysis (44.4%; n = 8) used HbA1c as the diagnostic tool for prediabetes, while 27.7% used FPG and another 27.7% employed mixed methods. This finding contrasts with the results reported by Nomura, Inoue & Akimoto (2012), who recommended using FPG or a two-step approach combining HbA1c and FPG for more accurate diabetes prediction. The preference for using HbA1c alone in most studies may reflect its convenience, particularly because it does not require fasting. However, our finding raises concerns about potential variability in accurately identifying individuals at risk.

In contrast to our study, which found a significantly higher prevalence of prediabetes among Saudi males (34.8%) compared to females (18.7%), meta-analyses studies from Bangladesh and Malaysia (Akhtar et al., 2020; Akhtar et al., 2022) reported no significant gender differences. Furthermore, a sensitivity analysis, excluding the Al Shehri et al. (2022) study from the male-only subgroup revealed a slight decrease in the pooled prevalence to 27.9% (95% CI [24.1%–32.1%]), yet it remained the highest among the subgroups. The higher prevalence among males suggests gender-related risk factors for prediabetes, including obesity, sedentary lifestyles, and unhealthy dietary habits. Supporting our findings, a cross-sectional study of 968 men and 2,029 women aged 30–75 in Riyadh found that men were 1.49 times more likely than women to report an unhealthy lifestyle. Men commonly cited a lack of enjoyment in physical activity, insufficient social support, and limited knowledge about healthy eating as key barriers. The authors concluded that younger men and older women are at higher risk for unhealthy lifestyle habits (Alquaiz et al., 2021). Likewise, a recent systematic review found that unhealthy dietary practices were widespread, largely driven by the taste and convenience of processed foods. Unhealthy eating and poor glycemic control were particularly associated with male gender, older age, and lower-income (Almutairi et al., 2023).

The gender-based difference observed in this meta-analysis—specifically, the higher prevalence of prediabetes among males—aligns with findings from a recent study involving 3,928 Saudi adults, where 67% consumed soft drinks and 30% consumed energy drinks weekly (Aljaadi et al., 2023). These secondary findings provide valuable context for understanding behavioral risk factors associated with sugary drink consumption in Saudi Arabia. The higher consumption observed among males, younger adults, and individuals with lower income may suggest greater susceptibility to marketing, peer influence, and lifestyle factors. Interestingly, an inverse association was found between energy drink intake and overweight/obesity. In addition, education level was not associated with soft or energy drink consumption. These insights highlight the need for gender-specific public health strategies to address prediabetes. Saudi males may benefit from targeted interventions such as lifestyle modification programs, early screening, or tailored education.

Rapid urban growth, infrastructure development, and societal transitions have significantly improved living standards in Saudi Arabia’s urban centers (Almulhim & Cobbinah, 2023). Almost all studies included in this meta-analysis were conducted in urban settings, which may have contributed to the high prevalence of prediabetes observed across different regions. The Central region reported the highest pooled prevalence of prediabetes at 26.0% (95% CI [17.5%–36.8%]) with a prediction interval of 5.1%–69.7%, indicating high variability in prevalence across studies. However, a sensitivity analysis for the Central region, excluding the Al Shehri et al. (2022) study, revealed a lower pooled prevalence of 22.8% (95% CI [20.8%–24.9%]) with a prediction interval of 17.1%–29.7%, indicating moderate variability across studies.

The significant impact of the Al Shehri et al. study on the pooled prevalence in the Central region may be related to specific characteristics of the study population. The participants in the Al Shehri et al. (2022) study were young males (aged 18–30 years) applying for military colleges. The high reported prevalence of prediabetes (55%) among this group may be attributed to lifestyle changes during recruitment preparation, such as altered dietary habits, reduced physical activity, and increased stress. The variation observed in the regional subgroup analysis highlights the need for further investigation into specific factors that may influence prediabetes prevalence. The highest pooled prevalence of prediabetes in the Central and Western regions may call for urgent interventions targeting lifestyle factors and healthcare availability or resources. Conversely, the lower pooled prevalence of prediabetes in the Eastern region may be attributed to the inclusion of only the Latif et al. study (Latif & Rafique, 2020). Consistent with findings among Saudis, Akhtar et al. (2022) observed variability in prediabetes prevalence across different states in Malaysia, suggesting that regional factors and socioeconomic diversity play a role. Similarly, Yitbarek et al. (2021) highlighted substantial heterogeneity in diabetes prevalence across regions within Ethiopia, especially among urban populations.

A sensitivity analysis based on study quality showed similar pooled prevalence estimates for high- and medium-quality studies (24.2% vs. 23.9%). The close alignment between these estimates suggests that study quality had minimal influence on the overall findings, reinforcing the robustness and reliability of the pooled prevalence reported in this meta-analysis (Fig. S7).

According to the GRADE methodology, the overall quality of evidence was considered moderate. Although study methodologies were sound and findings broadly consistent, the high heterogeneity observed among studies necessitated a cautious interpretation (Table S7).

The meta-regression analysis revealed an upward trend in prediabetes prevalence over time; however, this finding should be interpreted cautiously due to a Q-value of 1.43 (p = 0.2312). Additional factors, such as changes in diagnostic criteria and regional differences, may also contribute to the variability in prevalence. Lopez-Ruiz, Blazquez & Hasanov (2019) estimated that the Eastern Province, Riyadh, and Makkah together account for approximately 60% of Saudi Arabia’s gross domestic product (GDP). These may reflect differences in economic development, healthcare access, and early-life risk exposures. For example, Al-Hazzaa et al. (2012) examined lifestyle factors among adolescents in three major Saudi cities—Riyadh, Jeddah, and Al-Khobar—and found that lower physical activity levels and unhealthy dietary habits were significantly associated with overweight and obesity. Similarly, Alquaiz et al. (2021) found that men in Riyadh were more likely to engage in unhealthy lifestyle behaviors.

Overall, publication year did not significantly predict prediabetes prevalence in this meta-analysis. In contrast to our findings, Akhtar et al. (2020) reported an increase in prediabetes prevalence over time. They observed a progressive rise in diabetes prevalence in Bangladesh over recent decades. This may reflect increased longevity and possibly improved detection and diagnostic capabilities (Sanyaolu et al., 2023; Chan et al., 2025). Additionally, Akhtar et al. (2022) reported an increase in prediabetes prevalence in Malaysia between 1995–2010 and 2011–2020, suggesting that lifestyle shifts and dietary changes over time may have contributed.

Study Strengths

To the best of our knowledge, there are no national data are available estimating the pooled prevalence of prediabetes among adult Saudis in Saudi Arabia. This paper is the first systematic review and meta-analysis to synthesize data on the prevalence of prediabetes among adult Saudis living in Saudi Arabia. The strengths of this study include the use of a predefined and registered protocol and a detailed search strategy. Additionally, rigorous methodological and statistical approaches were applied, with two or more independent investigators involved at each stage of the review process.

We believe that these strengths enhance the reliability of the pooled prevalence estimate of prediabetes among adult Saudis in Saudi Arabia.

Study Limitations

Despite the strengths of this meta-analysis, several limitations should be acknowledged. First, the high level of heterogeneity (I2 = 99.1%) among the included studies may reflect significant variation in study populations, methodologies, and diagnostic approaches. As a result, the pooled prevalence estimate from this review should be interpreted with caution. Second, most included studies were conducted in urban settings, and no eligible studies specifically from the Northern region of Saudi Arabia were available. This may also limit the national representativeness of the meta-analysis findings. Third, the exclusion of non-English studies may have introduced language bias. However, this is unlikely to have significantly impacted the findings, as most academic research in Saudi Arabia is published in English. Lastly, selection bias in some studies—such as those recruiting from student populations, healthcare settings, or specific occupational groups—may limit the generalizability of their prevalence estimates.

Despite these limitations, this study offers the first comprehensive national estimate of prediabetes prevalence among Saudi adults. It highlights a significant public health concern and provides valuable guidance for healthcare planners and future research focused on diabetes prevention and early intervention.

Conclusion

Findings from the current meta-analysis show a high pooled prevalence of prediabetes among the adult Saudi population, with a particularly higher prevalence among males. The outcomes of this study emphasize the need for early lifestyle interventions, optimized screening programs, and effective resource allocation, all of which could help prevent the progression to type 2 diabetes.

Recommendations

In light of the high heterogeneity, we recommend that future population-based epidemiological studies standardize diagnostic criteria according to approved ADA or WHO guidelines and place greater focus on population sampling. Researchers should employ sampling methods that fairly represent all regions, ages, and genders, with particular attention to the underrepresented Northern province. Furthermore, clear reporting of study settings, diagnostic methods, and population details is essential for accurate interpretation and future meta-analysis. Together, these steps could reduce variability in prevalence estimates and improve comparability across studies.

Supplemental Information

Supplemental Information 1 PRISMA checklist

Supplemental Information 2 Supplementary Materials

Abbreviations

ADA American Diabetes Association

CI Confidence interval

DM Diabetes mellitus

FPG Fasting Plasma Glucose

HbA1c Glycated hemoglobin

IFG Impaired Fasting Glucose

JBI Joanna Briggs Institute

PRISMA Preferred reporting items for systematic reviews and meta-analyses

T2DM Type 2 diabetes mellitus

WHO World Health Organization

Additional Information and Declarations

Competing Interests

Author Contributions

Data Availability

The authors declare there are no competing interests.

Abdelmarouf Mohieldein conceived and designed the experiments, analyzed the data, prepared figures and/or tables, authored or reviewed drafts of the article, and approved the final draft.

Habeeb Ali Baig conceived and designed the experiments, performed the experiments, authored or reviewed drafts of the article, and approved the final draft.

Mahmoud Elhabiby conceived and designed the experiments, performed the experiments, authored or reviewed drafts of the article, and approved the final draft.

Abdullah Almushawwah conceived and designed the experiments, performed the experiments, authored or reviewed drafts of the article, and approved the final draft.

Mohamed Salih Mahfouz conceived and designed the experiments, analyzed the data, prepared figures and/or tables, authored or reviewed drafts of the article, and approved the final draft.

Atif H. Khirelsied conceived and designed the experiments, authored or reviewed drafts of the article, and approved the final draft.

Nour Abdelmarouf conceived and designed the experiments, performed the experiments, authored or reviewed drafts of the article, and approved the final draft.

GadAllah Modawe performed the experiments, authored or reviewed drafts of the article, and approved the final draft.

The following information was supplied regarding data availability:

This is a systematic review/meta-analysis.

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
