# Peer review of "Prediabetes in adult Saudis: a systematic review & meta-analysis of prevalence studies (2000–2024)"

_PeerJ, doi:10.7717/peerj.19778_

## Round 0.1 · original submission · Major Revisions

· Academic Editor

Major Revisions

Please address the reviewers' comments.

**Language Note:** The review process has identified that the English language must be improved. PeerJ can provide language editing services - please contact us at [email protected] for pricing (be sure to provide your manuscript number and title). Alternatively, you should make your own arrangements to improve the language quality and provide details in your response letter. – PeerJ Staff

Reviewer 1 ·

Basic reporting

This is a very relevant study. The authors have presented well in the English language.

The background provided for the study is great. Adding the background information on the prevalence of prediabetes before the study period could add value to what has changed over the years.


Abstract: Kindly follow and report using the PRISMA 2020 Abstract Checklist for the abstract.

Experimental design

Material and methods:
Search strategy
Line 105: Kindly clarify the PICO for easier understanding of the participants.
Line 106: Kindly provide the dates that were used to search each database. Please also mention if the same dates or different dates were used for all databases.
Inclusion and exclusion of studies:
Line 130: Can only including an English language paper miss important papers?
Line 130: It would be nice to understand why the January 2000 and September 2024 are chosen for the published paper search.

Data extractions: It would be nice to mention the inter-rater reliability if this was done, as there are multiple authors involved in data extractions and rating the quality of the studies.
Appraisal of the quality of included studies:
Line 155-156: Kindly mention how the conflicts were resolved
Also, presenting the Grading of Recommendations Assessment, Development and Evaluation (GRADE) methodology to summarise the quality of the retrieved studies could make your study more robust.

Discussion:
Line 306-309: Please check the font size. They do not appear the same
Line 379-383: It would be nice to present the secondary outcomes under the results sections and add these on the discussion to compare and contrast.

Strength and limitations:
Line 428: for the high heterogeneity, it is advisable to take caution in interpretation, that should be specifically mentioned for the readers.
Kindly also mention the implications of this study.

Supplementary Information:
Line 447: Please mention the specific supplementary files that are added

Abbreviations:
Please arrange alphabetically

Validity of the findings

The conclusion is well stated, but provided that the heterogeneity of the studies was very high, what other recommendations could be added?

·

Basic reporting

1. Clarity and Professional Language
While the manuscript is generally well-written, there are instances where the language could be more concise and professional. For example, in the Introduction section, some sentences are overly complex and could be simplified for better readability. Specifically, the sentence:
"Saudi Arabia has experienced extensive changes in lifestyle patterns. These changes include urbanization, shifts in dietary habits, and increased sedentary behavior—all of which are risk factors for prediabetes and diabetes (Munawir Alhejely et al., 2023)."
could be rephrased for clarity. Consider breaking it into shorter sentences or restructuring for better flow.

Suggestion: Please review the manuscript for clarity and ensure that the language is concise and unambiguous, especially in the introduction and discussion sections. Consider having a native English speaker or professional editor review the text for language improvements.

2. Figures and Tables
The figures and tables are generally well-presented, but there are areas where additional clarity could be beneficial. For instance, in Figure 2, the forest plot is informative, but the confidence intervals for some studies are quite wide (e.g., Al Shehri et al., 2022). This could be further explained in the text to help readers understand why these intervals are so broad and whether this affects the overall interpretation of the results.

Suggestion: Please provide a brief explanation in the discussion section regarding the wide confidence intervals observed in some studies and how this might impact the robustness of the pooled prevalence estimate.

3. Raw Data Availability
The manuscript mentions that raw data are provided in the supplementary files, but it is not entirely clear how accessible or well-organized these data are. For example, the supplementary files should include clear metadata identifiers and descriptions to ensure that future researchers can easily interpret and use the data.

Suggestion: Please ensure that the supplementary files are well-organized and include detailed metadata descriptions for each dataset. This will enhance the reproducibility and usability of your research.

Experimental design

1.Consistency of Diagnostic Criteria:

The article mentions the use of ADA and WHO diagnostic criteria but does not provide detailed information on whether there are differences in how these criteria were applied across studies. It is recommended to add an explanation in the methods section on how data from different diagnostic criteria were standardized.

2.Sensitivity of Diagnostic Methods:

The article mentions differences in results between FBG and HbA1c diagnostic methods but does not discuss the impact of these differences on the findings. It is recommended to elaborate on the influence of different diagnostic methods on the results in the discussion section.

Validity of the findings

1.Explanation of Heterogeneity:

The heterogeneity is high (I² = 99.1%). It is suggested to further explore the potential sources of heterogeneity in the discussion section, such as differences in lifestyle across regions or sample selection bias.

Additional comments

1.Study Limitations:

The article does not provide a detailed discussion of the study's limitations. It is suggested to add a discussion of the limitations in the discussion section, such as the timeliness of the data or potential sample selection bias.

Reviewer 3 ·

Basic reporting

This paper meets the standard of basic reporting.

Experimental design

The methodology and design are rigorous and well-reasoned. The authors may consider conducting a sensitivity analysis based on the quality of the studies.

Validity of the findings

The authors may consider further investigating why the prevalence of prediabetes in this study is significantly higher than in previous studies. They suggest that lifestyle is the main contributing factor. If lifestyle is indeed the primary reason, is the prevalence of diabetes in the Saudi population also higher than in other populations?

---

## Round 0.2 · Minor Revisions

· Academic Editor

Minor Revisions

Reviewer 1 ·

Basic reporting

Clear with the improved English version. Some references can still be added as suggested in specific headings.
If the PRISMA 2020 Abstract Checklist was used, it would be nice to add it to the supplementary file.

Experimental design

no comment

Validity of the findings

Discussion:
Line 382-391: Please check the font size of the in-text citation. They do not still appear the same.

Line 524-525: The additional factors can be elaborated with examples (diagnostic criteria changed example, WHO or ADA, and references), the authors have nicely raised the point on regional variation, which is very important; it would be nice to elaborate with references.

Line 528-529: It would be nice to add references to the point the authors have importantly raised regarding “the increased longevity and improved detection”.

Additional comments

Congratulations on the much-improved version.
If the PRISMA 2020 Abstract Checklist was used, it would be nice to add it to the supplementary file.

---

## Round 0.3 · accepted · Accept

· Academic Editor

Accept

Dear Dr. Mohieldein,

Thank you for submitting the revised version of your manuscript. After a thorough evaluation of your revisions by Reviewers and me, I am pleased to inform you that all reviewer comments have been satisfactorily addressed. Accordingly, your manuscript is now accepted for publication in PeerJ.

Sincerely,
Stefano Menini

·

Basic reporting

The authors have made substantial revisions addressing the previous concerns. The study design is now more robust, the data analysis is rigorous, and the results are clearly presented.

Experimental design

The authors have made substantial revisions addressing the previous concerns. The study design is now more robust, the data analysis is rigorous, and the results are clearly presented.

Validity of the findings

The authors have made substantial revisions addressing the previous concerns. The study design is now more robust, the data analysis is rigorous, and the results are clearly presented.